# Phosphorylated lipid-conjugated oligonucleotide selectively anchors on cell membranes with high alkaline phosphatase expression

Cheng Jin[1,2], Jiaxuan He[1], Jianmei Zou[1], Wenjing Xuan[1], Ting Fu[1], Ruowen Wang[2] & Weihong Tan[1,2,3]

Attachment of lipid tails to oligonucleotides has emerged as a powerful technology in constructing cell membrane-anchorable nucleic acid-based probes. In practice, however, conventional lipid-conjugated oligonucleotides fail to distinguish among different cell membranes. Herein, a phosphorylated lipid-conjugated oligonucleotide (DNA-lipid-P) is reported for alkaline phosphatase (ALP)-dependent cell membrane adhesion. In the absence of ALP, DNA-lipid-P with its poor hydrophobicity shows only weak interaction with cell membrane. However, in the presence of the highly expressed plasma membrane-associated ALP, DNA-lipid-P is converted to lipid-conjugated oligonucleotide (DNA-lipid) by enzymatic dephosphorylation. As a result of such conversion, the generated DNA-lipid has greater hydrophobicity than DNA-lipid-P and is thus able to insert into cell membranes in situ. Accordingly, DNA-lipid-P enables selective anchoring on cell membranes with elevated ALP level. Since elevated ALP level is a critical index of some diseases and even cancers, DNA-lipid-P holds promise for cell membrane engineering and disease diagnostics at the molecular level.

[1] Molecular Science and Biomedicine Laboratory (MBL), State Key Laboratory of Chemo/Biosensing and Chemometrics, College of Chemistry and Chemical Engineering, College of Life Sciences, Aptamer Engineering Center of Hunan Province, Hunan University, 410082 Changsha, Hunan, China. [2] Institute of Molecular Medicine (IMM), Renji Hospital, Shanghai Jiao Tong University School of Medicine, and College of Chemistry and Chemical Engineering, Shanghai Jiao Tong University, 200240 Shanghai, China. [3] Department of Chemistry and Department of Physiology and Functional Genomics, Center for Research at the Bio/Nano Interface, Health Cancer Center, UF Genetics Institute and McKnight Brain Institute, University of Florida, Gainesville, FL 32611-7200, USA. Correspondence and requests for materials should be addressed to W.T. (email: tan@chem.ufl.edu)

The mammalian cell membrane is a phospholipid bilayer structure which contains a variety of biological molecules, notably lipids and proteins[1,2]. Generally, the negatively charged moieties are on the cell membrane surface, while the hydrophobic alkyl chains are in the interior. Taking advantage of these features, lipid-conjugated oligonucleotides have been developed to anchor on the cell membrane through hydrophobic interactions between lipid tail and the hydrophobic area of cell membrane[3,4]. This has allowed the engineering of a variety of functional nucleic acids, such as DNAzymes and aptamers, on the cell membrane as biosensors or targeting ligands. For example, metal ion-responsive DNAzyme probes were immobilized on the cell membrane to monitor the changes of metal ions in cellular microenvironments[5]. Also, a cell-specific aptamer selected by systematic evolution of ligands by exponential enrichment (SELEX) was anchored on the cell membrane of T-lymphocytes in order to construct an aptamer-mediated cell recognition system for immunotherapy[6]. Recently, a membrane-anchored DNA probe was used to monitor dynamic and transient molecular encounters on the live cell membrane[7]. Finally, lipid-conjugated oligonucleotides were anchored on cell membranes to reconstitute three-dimensional microtissue arrays by layer-by-layer DNA-programmed assembly of cells[4].

In spite of these advances, conventional lipid-conjugated oligonucleotides suffer from poor selectivity among different cell membranes. This can be attributed to the following two reasons: (1) most mammalian cell membranes share similar phospholipid bilayer structures; and (2) the interaction between lipid tails and cell membranes is a nonselective physical interaction. To overcome the lack of selectivity caused by physical hydrophobic interaction between conventional lipid-conjugated oligonucleotides and different cell membranes, novel lipid tails with molecular recognition capability are needed. Therefore, since strong hydrophobic interaction between lipid-conjugated oligonucleotides and cell membranes is essential for their insertion into the cell membrane, we reasoned that the hydrophobicity of lipid-conjugated oligonucleotide would increase, but only after some type of selective reaction has taken place, which would, in turn, result in anchorage of lipid-conjugated oligonucleotides on the cell membrane of interest in situ.

Carrying this idea forward, we note that ALP is a universal hydrolyase responsible for the cleavage of phosphate groups from proteins and nucleic acids[8,9]. Elevated ALP levels occur in diabetes, bone diseases and even human cancers[10,11]. Because of its highly catalytic activity, ALP has been widely employed to convert phosphorylated hydrophilic probes to dephosphorylated hydrophobic molecules. The generated hydrophobic molecules can then self-assemble into supramolecular nanostructures for molecular imaging and cancer cell therapy[12–16]. Remarkably, ALP is highly expressed on the cell membranes of many cancer cells[14,17,18]. Inspired by the aforementioned pioneering works, we develop a phosphorylated lipid-conjugated oligonucleotide able to selectively anchor on the cell membranes with high ALP expression.

To accomplish this, we design and synthesize a DNA-lipid-P construct which has two negatively charged phosphate groups at the lipid terminus. As shown in Fig. 1, DNA-lipid-P has three components, as follows: (1) oligonucleotide (DNA); (2) N,N-didodecylaniline, acting as a hydrophobic lipid tail to insert into a cell membrane; and (3) two phosphate groups, acting as substrate of ALP that enables enzymatic cleavage on the cell membrane surface. Based on its poor hydrophobicity, DNA-lipid-P demonstrates weak anchoring on the cell membrane. However, after enzymatic dephosphorylation of DNA-lipid-P in the presence of high levels of ALP on the cell membrane surface, DNA-lipid-P converts to DNA-lipid with greater hydrophobicity.

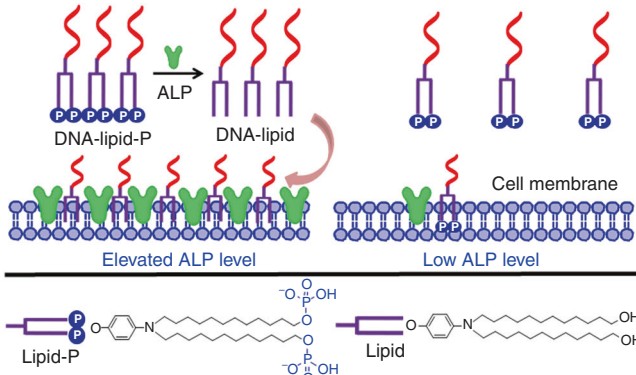

**Fig. 1** Schematic of ALP-dependent cell membrane anchorage of DNA-lipid-P

Such converted DNA-lipid with enhanced hydrophobicity can then selectively anchor on the cell membrane in situ.

## Results

**Hydrophobicity-dependent cell membrane anchorage**. Conventional lipid-conjugated oligonucleotides are amphiphilic molecules composed of two segments: negatively charged oligonucleotides as the hydrophilic moieties and lipids as the hydrophobic groups[19]. In order to generate a variety of lipid-conjugated oligonucleotides with different hydrophobicities, lipid tails with different alkyl chain lengths were synthesized (Supplementary Fig. 1) and conjugated with DNA (Fig. 2a and Supplementary Fig. 3). High-performance liquid chromatography (HPLC) is a universal tool to evaluate the hydrophobicity of oligonucleotides by comparing their retention time. A greater retention time indicates stronger hydrophobicity of oligonucleotides[20]. As shown in Fig. 2b, the retention times of C6-DNA, C9-DNA, C12-DNA, and C15-DNA are 19.2, 23.8, 27.4, and 30.8 min, respectively, demonstrating that the longer alkyl chains, i.e., with stronger hydrophobicity, in the lipid tail provide stronger hydrophobicity of lipid-conjugated oligonucleotides.

Next, flow cytometry was used to investigate the binding affinity of C6-DNA, C9-DNA, C12-DNA, and C15-DNA to the cell membrane. As shown in Fig. 2c, DNA-treated HepG2 cells show the weakest fluorescence intensity, whereas C15-DNA-treated HepG2 cells show the strongest fluorescence intensity. In addition, confocal fluorescence microscopy imaging of HepG2 cells further indicates that almost all fluorescence signals were located on the cell membrane (Fig. 2d). This cumulative evidence shows that the adhesion of lipid-conjugated oligonucleotides to the cell membrane primarily depends on the hydrophobicity of the lipid tail. In other words, lipid-conjugated oligonucleotides with stronger hydrophobicity have tighter anchorage on the cell membrane.

**Synthesis of DNA-lipid-P**. Next, we attempted to generate DNA-lipid-P through DNA synthesis and phosphoramidite chemistry. However, conventional lipid phosphoramidites lack 4,4′-dimethoxytrityl-(DMT-)protected hydroxyl at the terminus of alkyl chains for further functionalization. Therefore, to synthesize DNA-lipid-P, a novel lipid phosphoramidite was developed. As shown in Fig. 3a, this lipid phosphoramidite has two DMT-protected hydroxyls at the terminus of alkyl chains which enable further chemical phosphorylation during solid-phase synthesis (Fig. 3b). As such, after iterative synthesis on a DNA synthesizer, DNA-lipid-P was generated with acceptable yield (about 38%) (Supplementary Fig. 7) and high purity (>99%) (Supplementary Table 3).

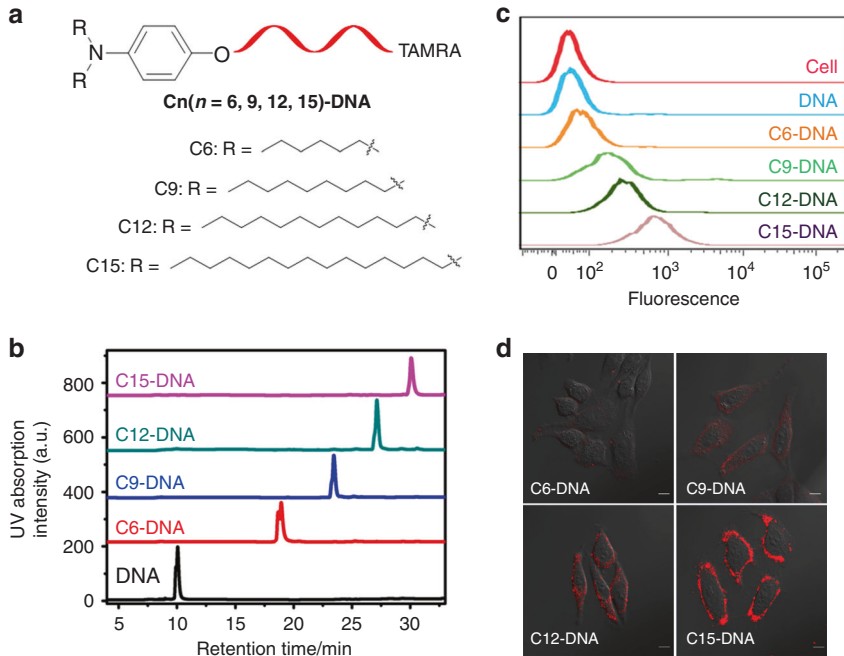

**Fig. 2** Hydrophobicity-dependent cell membrane adhesion of lipid-conjugated oligonucleotides. **a** Schematic illustration of the structure of C6-DNA, C9-DNA, C12-DNA and C15-DNA with different alkyl chain lengths in lipid tails. **b** HPLC chromatograms of DNA, C6-DNA, C9-DNA, C12-DNA and C15-DNA. **c** Flow cytometry of HepG2 cells treated with 1 μM TAMRA-labeled DNA, C6-DNA, C9-DNA, C12-DNA or C15-DNA. **d** Confocal fluorescence microscopy imaging of HepG2 cells treated with 1 μM TAMRA-labeled C6-DNA, C9-DNA, C12-DNA or C15-DNA. Scale bar is 10 μm

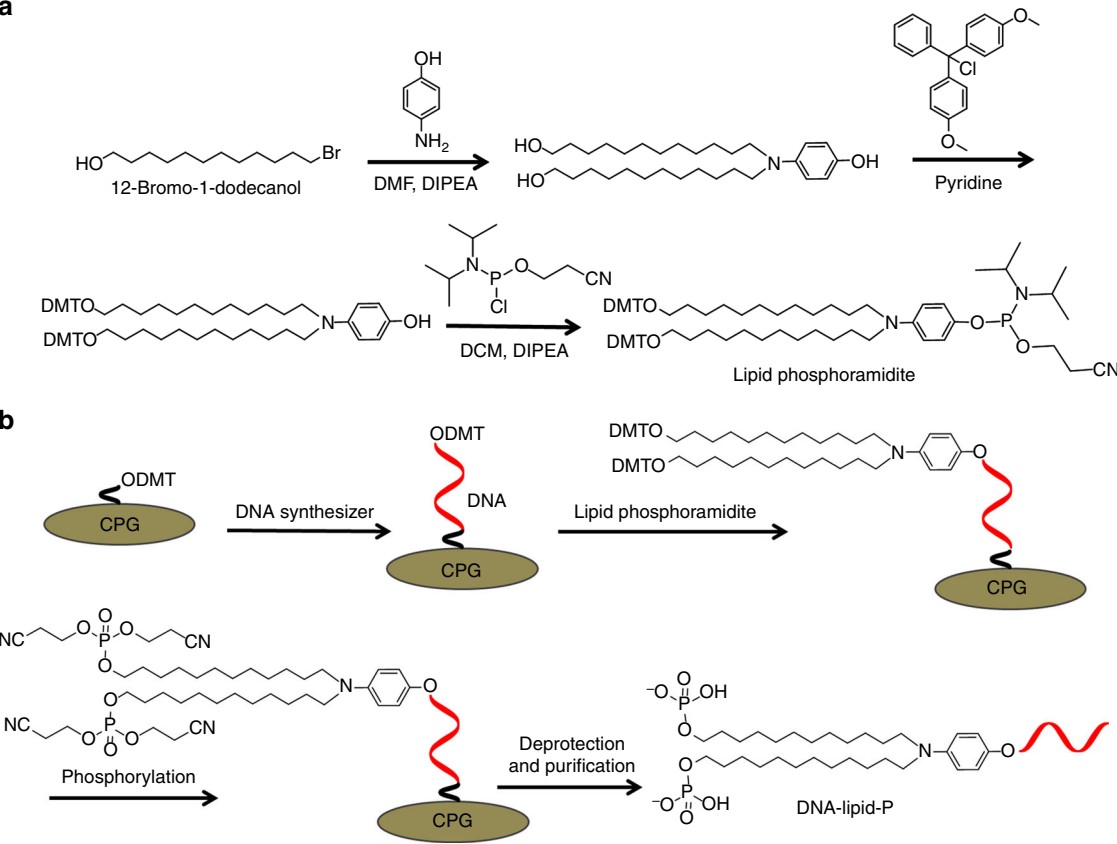

**Fig. 3** Synthesis of lipid phosphoramidite and DNA-lipid-P. **a** Synthesis route of lipid phosphoramidite. DMF indicates N,N-dimethylformamide. DIPEA indicates N,N-diisopropylethylamine. DCM indicates dichloromethane. **b** DNA synthesis route of DNA-lipid-P. Chemical phosphorylation was performed by coupling bis(2-cyanoethyl)-N,N-diisopropylphosphoramidite with oligonucleotide on a DNA synthesizer. CPG indicates controlled pore glass

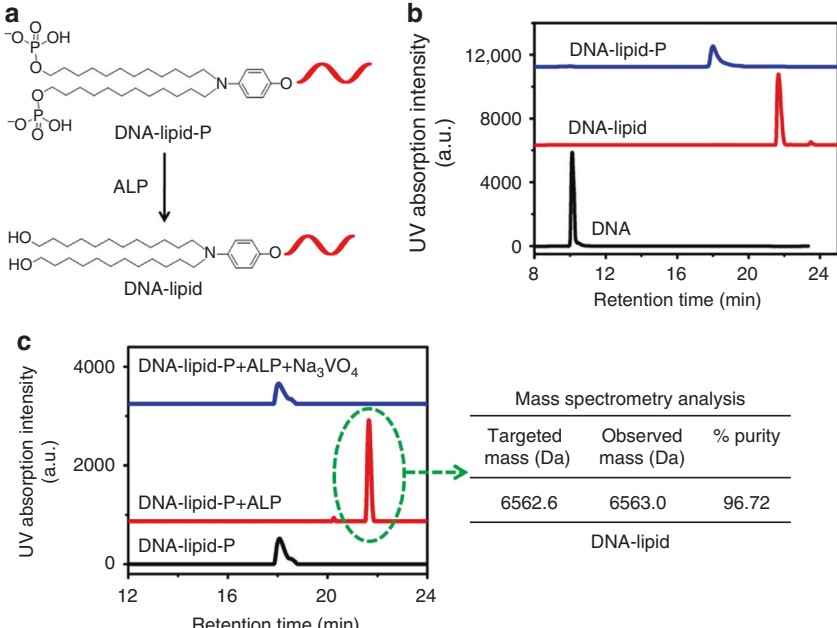

**Fig. 4** Enzymatic dephosphorylation of DNA-lipid-P. **a** Schematic illustration of ALP-induced dephosphorylation of DNA-lipid-P. **b** HPLC chromatograms of DNA, DNA-lipid and DNA-lipid-P. **c** Left: HPLC chromatograms of DNA-lipid-P in buffer solution (black line), DNA-lipid-P treated with ALP (0.5 U mL$^{-1}$) in buffer solution (red line) and DNA-lipid-P treated with ALP (0.5 U mL$^{-1}$) in buffer solution containing 2 mM Na$_3$VO$_4$ (blue line). Right: mass spectrometry analysis of DNA-lipid-P after treatment with ALP (0.5 U mL$^{-1}$)

**Enzymatic dephosphorylation of DNA-lipid-P.** Having confirmed the successful synthesis of DNA-lipid-P, we further investigated whether it could be converted to DNA-lipid by ALP (Fig. 4a). As shown in Fig. 4b, the retention times of DNA-lipid and DNA-lipid-P are 21.8 and 18.0 min, respectively, suggesting that the phosphorylation of lipid tail does, indeed, significantly decrease its hydrophobicity. Next, ALP was used to enzymatically dephosphorylate DNA-lipid-P. Upon incubation with ALP (0.5 U mL$^{-1}$), the DNA-lipid-P peak at 18.0 min disappeared; instead, a new DNA peak at 21.8 min was observed (Fig. 4c and Supplementary Fig. 8). The molecular weight of DNA at 21.8 min post-dephosphorylation was 6563.0 Da, which is consistent with the molecular weight of DNA-lipid (6562.6 Da), demonstrating that DNA-lipid-P had been converted to DNA-lipid (Fig. 4c and Supplementary Fig. 9). Sodium orthovanadate (Na$_3$VO$_4$), a competitive inhibitor of ALP, was used to inhibit ALP activity to provide more proof that only ALP is responsible for dephosphorylation[21]. As shown in Fig. 4c and Supplementary Fig. 10, Na$_3$VO$_4$ (2 mM) completely inhibited the conversion of DNA-lipid-P to DNA-lipid. In a word, ALP removed the phosphate groups from DNA-lipid-P by enzymatic hydrolysis, producing DNA-lipid with greater hydrophobicity than DNA-lipid-P, thereby promoting cell membrane anchorage.

**ALP-dependent cell membrane anchorage of DNA-lipid-P.** Encouraged by the enzymatic dephosphorylation of DNA-lipid-P in homogeneous buffer solution, we further investigated whether DNA-lipid-P would show ALP-dependent anchorage on the HepG2 cell membrane with high expression of ALP[16,22–24]. Flow cytometry of HepG2 cells treated with DNA-lipid, DNA-lipid-P, or DNA-lipid-P + Na$_3$VO$_4$ was performed, and the mean fluorescence intensity was recorded. As shown in Fig. 5b and Supplementary Fig. 11, DNA-lipid-P-treated HepG2 cells exhibit fluorescence intensity comparable with that of DNA-lipid-treated cells. However, poor fluorescence signals were observed in cells treated with DNA-lipid-P in Na$_3$VO$_4$-containing buffer solution (DNA-lipid-P + Na$_3$VO$_4$ group). In addition, most of

the fluorescence signal was located on the cell membrane, even 6 h post-incubation (Supplementary Fig. 13). These results demonstrate that ALP, which is highly expressed on the HepG2 cell membrane, induces the dephosphorylation of DNA-lipid-P and, subsequently, the anchorage of DNA-lipid on the cell membrane in situ.

To further evaluate whether the oligonucleotides had anchored on the external surface of cell membrane after incubation, Dabcyl-labeled complementary DNA (Dabcyl-cDNA) was designed to hybridize with the anchored DNA to quench its fluorescence (Supplementary Fig. 14)[7]. As shown in Fig. 5d and Supplementary Fig. 15, decreases of about 70 and 66% in fluorescence were observed in DNA-lipid and DNA-lipid-P groups, respectively, suggesting that oligonucleotides had, indeed, anchored on the external membrane of HepG2 cells. Moreover, neither DNA-lipid nor DNA-lipid-P was able to aggregate into micellar nanoparticles in buffer solution in any appreciable way, suggesting that both DNA-lipid and DNA-lipid-P interacted with cell membranes as monomers, not as aggregates (Supplementary Fig. 17). Taken together, it can be concluded that DNA-lipid-P anchors to the HepG2 cell membrane in an ALP-dependent manner.

**Selective cell membrane anchorage of DNA-lipid-P.** Next, DNA-lipid-P was employed to distinguish different cell membranes with different levels of ALP expression. HepG2 is a hepatoma cell line with elevated ALP level on the cell membrane, while U-2 OS is an osteosarcoma cell line with quite low expression of ALP on the cell membrane[25–28]. In consistent with the previous literatures, the measured ALP activity in HepG2 cells is much higher than that in U-2 OS cells with a multiple of about 145-fold (Fig. 6b, c and Supplementary Table 4). DNA-lipid alone fails to selectively anchor on either cell membrane. Meanwhile, DNA-lipid-P-treated HepG2 cells show obviously greater fluorescence intensity on the cell membrane compared with that of DNA-lipid-P-treated U-2 OS cells with ratios of fluorescence intensity in 1 μM concentration of about 1.9-fold (Fig. 6d, e and

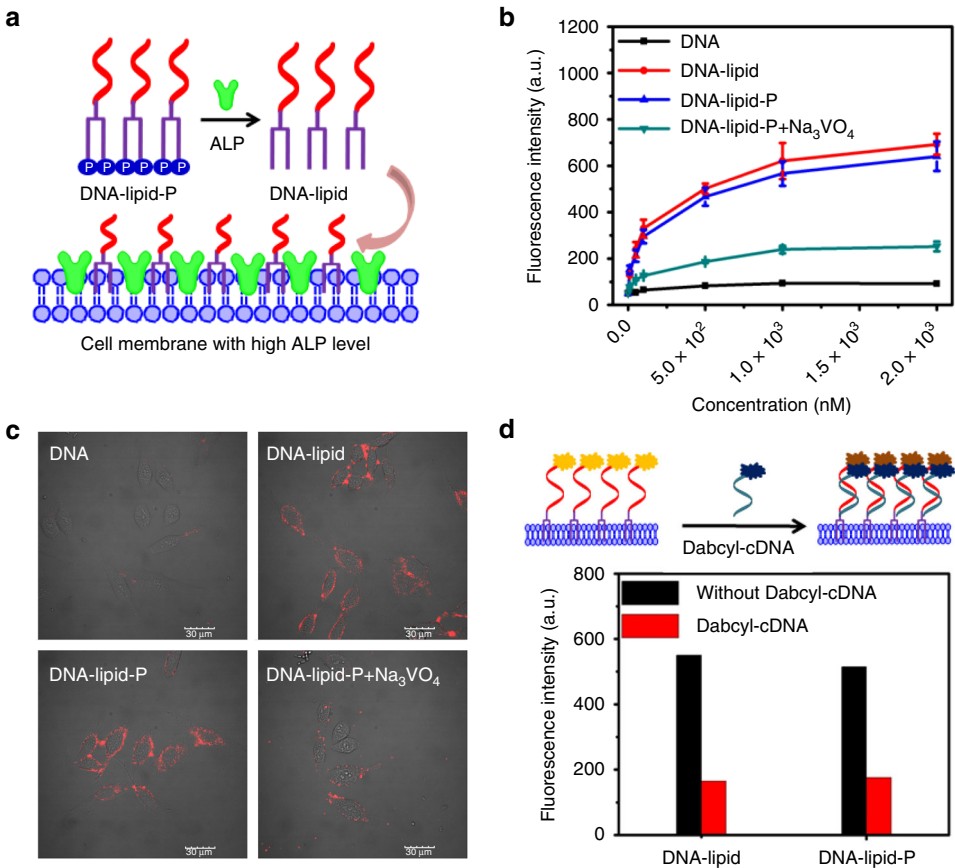

**Fig. 5** ALP-dependent cell membrane anchorage of DNA-lipid-P. **a** Schematic illustration of ALP-dependent cell membrane anchorage of DNA-lipid-P. **b** Mean fluorescence intensity of HepG2 cells treated with TAMRA-labeled DNA, DNA-lipid, DNA-lipid-P or DNA-lipid-P + Na$_3$VO$_4$ (2 mM) for one hour at 37 °C. Error bars indicate standard deviation. **c** Confocal fluorescence microscopy imaging of HepG2 cells treated with 1 μM TAMRA-labeled DNA, DNA-lipid, DNA-lipid-P or DNA-lipid-P + Na$_3$VO$_4$ (2 mM) for one hour at 37 °C. Scale bar is 30 μm. **d** Hybridization-induced fluorescence quenching of membrane-anchored TAMRA-labeled DNA-lipid or DNA-lipid-P with Dabcyl-cDNA

Supplementary Fig. 19), indicating that DNA-lipid-P can selectively anchor on the surface membrane of cells with elevated ALP levels.

## Discussion

In summary, a phosphorylated lipid-conjugated oligonucleotide for ALP-dependent cell membrane anchorage was developed by integrating enzymatic cleavage and DNA-based engineering of cell membranes. DNA-lipid-P could anchor on cell membranes expressing a high level of ALP by dephosphorylation-induced increase of hydrophobicity. In addition, DNA-lipid-P could be synthesized automatically on a DNA synthesizer with acceptable yields, well-defined molecular structure and high purity. Notably, few studies have reported on the functionalization of lipid tails in lipid-conjugated oligonucleotides. The strategy reported in this work provides a facile route toward such functionalization and synthesis and offers a model for the improvement of properties of conventional lipid-conjugated oligonucleotides.

## Methods

**Synthesis of lipid phosphoramidite.** Three grams 12-bromo-1-dodecanol (11.3 mmol), DIPEA (2.90 g, 22.6 mmol), and *p*-aminophenol (0.61 g, 5.6 mmol) were dissolved in 15 mL anhydrous DMF. The reaction was allowed to reflux under the protection of nitrogen gas and monitored by thin-layer chromatography (TLC). When the reaction was completed, the mixture was diluted with 100 mL dichloromethane and washed successively with saturated NaHCO$_3$ and brine. The organic layer was collected and dried by anhydrous Na$_2$SO$_4$. The solvent was removed by rotary evaporator. After purification through a flash chromatography column, 2.1 g

hydroxylated lipid was obtained (79% yield). ESI-MS calculated molecular weight is 477, and observed molecular weight is 478.

Two grams hydroxylated lipid (4.2 mmol) and 4,4′-dimethoxytrityl chloride (3.0 g, 8.8 mmol) were dissolved in 30 mL anhydrous pyridine. The reaction was allowed to stir overnight at room temperature under the protection of nitrogen gas. Then, the solvent was removed by rotary evaporator. After purification through a flash chromatography column, 2.18 g DMT-protected lipid was obtained as a colorless foamed solid (48% yield). $^1$H NMR (400 MHz, acetone-d6): δ 7.46 (d, J = 7.6 Hz, 4 H), 7.32 (d, J = 7.7 Hz, 8 H), 7.29 (d, J = 7.6 Hz, 4 H), 7.21 (t, J = 7.2 Hz, 2 H), 6.87 (d, J = 7.7 Hz, 8 H), 6.76 (s, 1 H), 6.69 (d, J = 8.1 Hz, 2 H), 6.62 (d, J = 7.9 Hz, 2 H), 3.78 (s, 12 H), 3.15 (t, J = 7.1 Hz, 4 H), 3.05 (t, J = 6.2 Hz, 4 H), 1.65–1.57 (m, 4 H), 1.51 (s, 4 H), 1.42–1.35 (m, 4 H), 1.28 (s, 28 H). ESI-MS calculated molecular weight is 1082, and observed molecular weight is 1082. The NMR spectra are shown in Supplementary Fig. 40.

DMT-protected lipid (2.10 g, 1.6 mmol) and DIPEA (0.63 g, 4.9 mmol) were dissolved in 30 mL anhydrous dichloromethane. The reaction bottle was allowed to cool in an ice bath under the protection of nitrogen gas. Then, 2-cyanoethyl N,N-diisopropylchlorophosphoramidite (0.58 g, 2.46 mmol) was added dropwise. The ice bath was removed, and the reaction was stirred for an additional 1 h. When the reaction was completed, 100 mL dichloromethane was added, and the mixture was washed successively by saturated NaHCO$_3$, brine, and water. The organic layer was collected and dried by anhydrous Na$_2$SO$_4$. The solvent was removed by rotary evaporator. After purification through a flash chromatography column, 1.70 g lipid phosphoramidite was obtained as a colorless solid (81% yield). $^1$H NMR (400 MHz, acetone-d6): δ 7.46 (d, J = 7.7 Hz, 4 H), 7.32 (d, J = 8.5 Hz, 8 H), 7.29 (d, J = 7.7 Hz, 4 H), 7.20 (t, J = 7.3 Hz, 2 H), 6.90 (d, J = 9.6 Hz, 2 H), 6.87 (d, J = 8.5 Hz, 8 H), 6.63 (d, J = 8.7 Hz, 2 H), 3.97–3.88 (m, 2 H), 3.77 (s, 14 H), 3.24 (t, J = 7.4 Hz, 4 H), 3.05 (t, J = 6.4 Hz, 4 H), 2.78 (t, J = 6.0 Hz, 2 H), 1.65–1.51 (m, 8 H), 1.39 (d, J = 6.9 Hz, 4 H), 1.30 (d, J = 19.3 Hz, 28 H), 1.19 (d, J = 6.3 Hz, 12 H). $^{31}$P NMR (162 MHz, acetone-d6): δ 146.4. $^{13}$C NMR (101 MHz, acetone-d6): δ 158.59, 145.79, 144.62, 136.58, 129.95, 128.07, 127.59, 126.49, 120.98, 120.90, 113.30, 112.88, 85.57, 63.02, 59.64, 59.16, 58.98, 54.59, 51.24, 43.36, 43.24, 29.84, 29.40, 29.24, 27.17, 26.93, 26.17, 24.07, 24.00, 23.94, 23.86, 19.94. ESI-MS calculated molecular weight

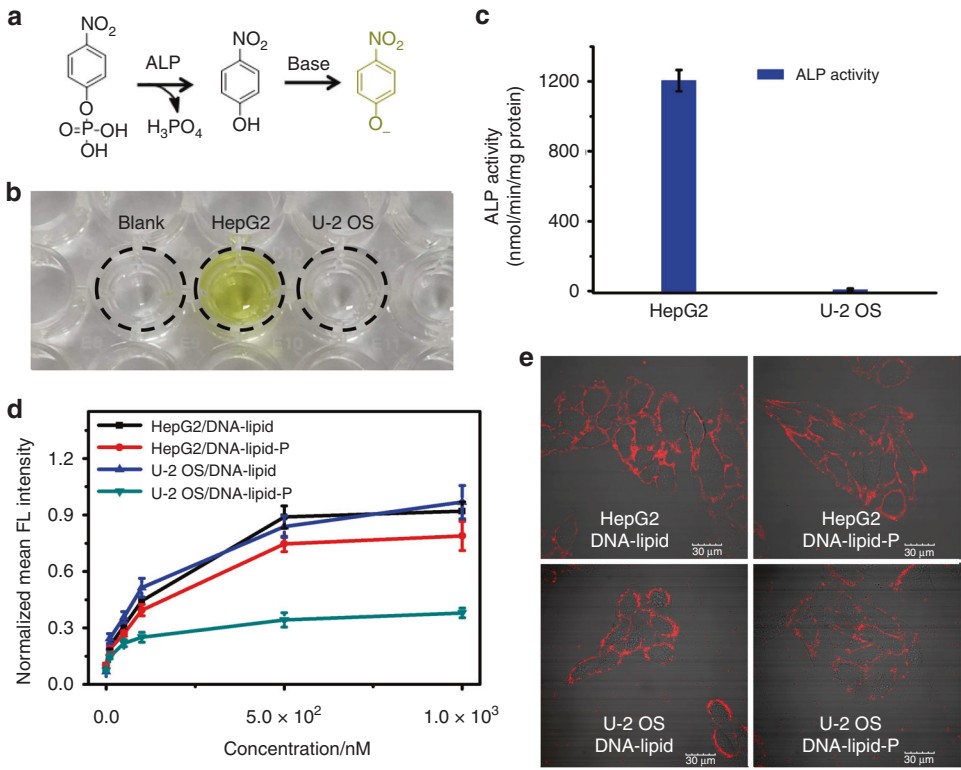

**Fig. 6** Selective cell membrane anchorage of DNA-lipid-P. **a** The reaction equation and chromogenic mechanism of *p*-nitrophenyl phosphate in the measurement of ALP enzymatic activity. **b** Photograph of *p*-nitrophenyl phosphate before (blank) and after enzymatic dephosphorylation in cell lysis-containing buffer solution. **c** Calculated enzymatic activity of ALP in HepG2 and U-2 OS cells. Error bars indicate standard deviation. **d** The normalized mean fluorescence intensity of HepG2 or U-2 OS cells treated with TAMRA-labeled DNA-lipid or DNA-lipid-P for one hour at 37 °C. Error bars indicate standard deviation. **e** Confocal fluorescence microscopy imaging of HepG2 and U-2 OS cells treated with 1 μM TAMRA-labeled DNA-lipid or DNA-lipid-P for one hour at 37 °C. Scale bar is 30 μm

is 1282, observed molecular weight is 1282. The NMR spectra are shown in Supplementary Figs. 41–43.

**DNA synthesis**. All DNA strands used in this work were synthesized on the PolyGen C12 DNA/RNA solid-phase synthesizer on a 0.1 micromolar scale, using the corresponding CPG. The labeled phosphoramidites were reacted with CPG for 600 s on the DNA synthesizer. After synthesis, the obtained oligonucleotides were cleaved and deprotected from the CPG, followed by precipitation in cold ethanol solution at −20 °C overnight. After centrifugation to remove the supernatant solution, oligonucleotides were dissolved with 0.1 M triethylamine acetate (TEAA) and purified by reversed phase HPLC (Agilent 1260 Infinity) using a BioBasic 4 column. Finally, the 4,4′-dimethoxytrityl group was removed from DNA by adding 80% acetic acid aqueous solution and precipitating in cold ethanol again. After drying in vacuum and desalting, the obtained oligonucleotides were quantified by measuring their absorbances at 260 nm.

**Mass spectrometry analysis of oligonucleotides**. After desalting using a NAP-5 column (GE Healthcare), the oligonucleotides (1 nmol) were sent to Sangon Biotechnology (Shanghai, China) as a powder sample for mass spectrometry analysis. Analytic conditions were as follows: instrument: Thermo LCQ, ESI-MS, negative ion mode; solvent condition: hexafluoroisopropanol aqueous solution (HFIPA-H$_2$O); scanned mass range set from 550 to 1550 (mass-to-charge ratio).

**Retention time analysis of oligonucleotides**. The corresponding oligonucleotides were diluted with TBS buffer (10 mM Tris-HCl buffer, pH 7.4, 137 mM NaCl, 4.7 mM KCl and 5 mM MgCl$_2$) to the final concentration of 20 μM (100 μL). Then, the above solution was injected into the HPLC for retention time analysis. The elution program was shown in Supplementary Table 2.

**Dephosphorylation of DNA-lipid-P in buffer solution**. DNA-lipid-P was diluted with TBS buffer to a final concentration of 20 μM (100 μL). Then, ALP was added to the above solution. After mixing, the solution was incubated at 37 °C for 10 min and then 75 °C for 5 min to deactivate ALP.

**Flow cytometry assay**. HepG2 and U-2 OS cells were purchased from Xiangya School of Medicine, Central South University, China. Cell lines were authenticated by short tandem repeat (STR)-profiling. HepG2 and U-2 OS cells were cultured in 6-well plates and maintained in DMEM medium supplemented with 10% fetal bovine serum and 0.5 mg mL$^{-1}$ penicillin-streptomycin at 37 °C in 5% CO$_2$. Then, cells were digested from 6-well plates by using 0.2% EDTA and washed with TBS buffer twice. Then, about 300 thousand cells were incubated with the corresponding oligonucleotides at 37 °C for 1 h. When incubation was completed, cells were washed twice with TBS buffer and subjected to flow cytometry. For cDNA hybridization experiments, TAMRA-labeled DNA-lipid and DNA-lipid-P were incubated with HepG2 cells for 1 h. After washing twice, cells were incubated with Dabcyl-cDNA in TBS buffer for 10 min. Then, cells were washed twice again and subjected to flow cytometry assay. Binding affinity of every group was measured three times with the same sample.

**Measurement of ALP enzymatic activity**. After washing with TBS buffer, HepG2 and U-2 OS cells were harvested in cell lysis buffer (20 mM Tris, 150 mM NaCl, 1% Triton X-100) (Cell lysis buffer for Western and IP without inhibitors, Beyotime Biotechnology), respectively. The harvested cells were sonicated and centrifuged at $19,314 \times g$ for 5 min. Then, the supernatant was collected for the assays of protein concentration and ALP activity. ALP activity was assessed at 37 °C using *p*-nitrophenyl phosphate as a substrate in 0.1 M diethanolamine (DEA)-HCl buffer (pH 9.8) containing 5 mM MgCl$_2$ (Alkaline Phosphatase Assay Kit, Beyotime Biotechnology). Protein concentrations were determined using a BCA Protein Assay Kit (Beyotime Biotechnology). ALP activity was expressed as nmol min$^{-1}$ mg$^{-1}$ protein. ALP enzymatic activity in HepG2 and U-2 OS cells was measured three times with the same sample.

**Confocal fluorescence microscopy imaging assay**. HepG2 and U-2 OS cells were placed in a 35 mm culture dish and cultured for 36 h before the experiment. Cells were washed twice with TBS buffer. Then, the corresponding oligonucleotides were incubated with cells in TBS buffer for the desired time. After incubation, cells were washed twice with TBS buffer and then subjected to confocal fluorescence microscopy imaging. For co-stain experiments with cell membrane probe (DiO), the corresponding oligonucleotides were incubated with HepG2 cells for the

desired time at 37 °C. Then, DiO (1 μM) was added and the cells were incubated for an additional 15 min. Cells were washed twice with TBS buffer and subjected to confocal fluorescence microscopy imaging assay. For cDNA hybridization experiments, TAMRA-labeled DNA-lipid-P was incubated with HepG2 cells for 1 h. After washing twice, cells were incubated with Dabcyl-cDNA in TBS buffer for 10 min. Then, cells were washed twice again and subjected to confocal fluorescence microscopy imaging assay.

**Cell viability assay**. The cytotoxicity of oligonucleotides to HepG2 and U-2 OS cells was evaluated using a 96-well proliferation assay. HepG2 and U-2 OS cells were placed in 96-well plates and grown to around 30% confluence for 24 h before the experiment. Then, oligonucleotides in DMEM culture medium (10% FBS) were added to the above 96-well plates. Cells were cultured for an additional 24 or 48 h at 37 °C. Finally, CCK-8 was added to each well, and the 96-well plates were subjected to absorption measurement at 450 nm using a microplate reader.

**Reporting summary**. Further information on research design is available in the Nature Research Reporting Summary linked to this article.

## Data availability
All data supporting the finding of this study are available with the Article and its Supplementary Information or from the corresponding author upon the reasonable request.

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

## Acknowledgements
This work is supported by China NSFC grants (NSFC 21521063 and NSFC 21327009) and by US NIH GM079359 and NSF 1645215.

## Author contributions
C.J. and W.T. conceived and designed the experiments; C.J. designed and performed the most experiments and data analyses; C.J., J.H. and W.X. performed the synthesis of phosphoramidites; C.J. and J.Z. performed the culture of cells; T.F. and R.W. participated the data analyses; C.J. and W.T. prepared the manuscript.

## Additional information

**Competing interests:** The authors declare no competing interests.

