## [Peer Review File · Nature Communications]

Reviewers' comments:

Reviewer #1 (Remarks to the Author):

This manuscript by Tan and coworkers demonstrate the selective anchoring of phosphorylated lipid-conjugated oligonucleotide on cell membranes with high expression of ALP. The authors designed and synthesized a DNA-lipid-P construct which has two negatively charged phosphate group at the lipid terminus. Due to the poor hydrophobicity, DNA-lipid-P exhibits weak anchoring on cell membrane, while it can anchor on cell membrane efficiently after enzymatic dephosphorylation. Therefore, I support the acceptance of this work after the authors revise their manuscript to address the following issues.

1. Will the DNA-lipid-P disrupt the membrane of cells after anchoring on the cell membrane? Or it will also anchor on the membrane of organelles? The authors need to provide the cell viability results of cells treated with DNA-lipid-P and DNA-lipid and the confocal images for longer incubation time.
2. The authors should provide the measured ALP activity on HepG2 and U-2OS cells.
3. Is the DNA-lipid-P be stable in cellular environment? Will it be degraded by nuclease? Discussion is needed on this point.

Reviewer #2 (Remarks to the Author):

In this manuscript, the authors reported a phosphorylated lipid-conjugated oligonucleotide (DNA-lipid-P) for ALP-dependent cell membrane anchorage. DNA-lipid-P with poor hydrophobicity shows very weak interaction with cell membrane in the absence of ALP. However, in the presence of ALP, hydrophilic DNA-lipid-P can be converted to hydrophobic DNA-lipid after enzymatic cleavage, resulting in strong fluorescence in the membrane of ALP highly expressed cancer cells. The novelty of this work is high and the manuscript is well prepared. The authors also provide a facile route toward the functionalization and a model to improve the properties of conventional lipid-conjugated oligonucleotides. This work is recommended for publication in Nature Communications after careful revisions.

- 1) "tcell" in Page 4 Line 1 should be checked out and revised.
- 2) Compound characterization: The coupling constant should be given in ¹H-NMR data in the synthesis part. The H atoms of ¹H-NMR data (Supplementary Figure 35) of lipid phosphoramidite are not correct and should be carefully checked out.
- 3) For better understanding, the chemical structures of TAMRA and Dabcyl should be added in the Supplementary Information.
- 4) I suggest the authors to carry out co-stain experiment with commercial membrane probe to directly verify the location of the probe DNA-lipid-P.
- 5) Direct imaging data should also be added to evaluate the phenomenon that oligonucleotides anchor on the external surface of cell membrane by using Dabcyl-labeled complementary DNA.
- 6) The authors should use TAMRA not Nile Red (Supplementary Figure 13) to demonstrate that neither DNA-lipid nor DNA-lipid-P is able to aggregate into micellar nanoparticles in buffer solution. Direct evidence like Dynamic Light Scattering (DLS) data should be also provided.
- 7) Since elevated ALP level is a critical index of some diseases and even cancers, so if the authors could provide further imaging data with normal cell line which shows low ALP expression, I believe the readers can have a comprehensive appreciation of this work
- 8) The abbreviation for "Ange. Chem., Int. Ed." (Ref. 6, 16, 17 and 19) is not correct, please revise that. The name of the journal in Ref. 22 should also be checked carefully.

Response to the Reviewers' Comments

Reviewers' comments:

Reviewer #1 (Remarks to the Author):

This manuscript by Tan and coworkers demonstrate the selective anchoring of phosphorylated lipid-conjugated oligonucleotide on cell membranes with high expression of ALP. The authors designed and synthesized a DNA-lipid-P construct which has two negatively charged phosphate group at the lipid terminus. Due to the poor hydrophobicity, DNA-lipid-P exhibits weak anchoring on cell membrane, while it can anchors on cell membrane efficiently after enzymatic dephosphorylation. Therefore, I support the acceptance of this work after the authors revise their manuscript to address the following issues.

1. Will the DNA-lipid-P disrupt the membrane of cells after anchoring on the cell membrane? Or it will also anchor on the membrane of organelles? The authors need to provide the cell viability results of cells treated with DNA-lipid-P and DNA-lipid and the confocal images for longer incubation time.

Response: Understand the referee's concern and thanks for the careful review. According to the reviewer's suggestions, cell cytotoxicity of DNA-lipid-P and DNA-lipid (concentration range: from 0.1 nM to 1 μ M) to HepG2 and U-2 OS cells was measured. As shown in Supplementary Figure 19, no obvious decline of cell viability was observed in all cases even after 48 hours post-incubation, possibly demonstrating neither DNA-lipid-P nor DNA-lipid disrupt the cell membrane.

We also notice that a conjugate of tyrosine and cholesterol (TC), formed by enzyme catalyzed dephosphorylation of phosphorylate TC (pTC), self-assembles selectively on or in cancer cells (Bing Xu et al., J. Am. Chem. Soc. 2018, 140, 9566-9573). Acting as polypharmaceuticals, the assemblies of TC augment lipid rafts, aggregate extrinsic cell death receptors (e.g., DR5, CD95, or TRAILR), modulate the expression of oncoproteins (e.g., Src and Akt), disrupt the dynamics of cytoskeletons (e.g., actin filaments or microtubules), induce ER stress, and increase the production of reactive oxygen species (ROS), thus resulting in cell death and preventing acquired drug resistance (Bing Xu et al., doi: 10.1158/1541-7786.MCR-18-0931). However, in our work, neither DNA-lipid-P nor DNA-lipid shows appreciable cytotoxicity to HepG2 and U-2 OS cells. The reason possibly can be attributed to the following two points: (1) the concentration of DNA-lipid-P (1 μ M) is much lower than that of pTC in the previous paper (IC₅₀: about dozens of μ M) (J. Am. Chem. Soc. 2018, 140, 9566-9573); and (2) the negatively charged oligonucleotides prevent DNA-lipid-P from internalizing into cytoplasm after enzymatic dephosphorylation, but anchor on cell membrane (Supplementary Figure 12).

As to the comment whether DNA-lipid-P anchor on the membrane of organelles. Commercial cell membrane probe, 3,3'-dioctadecyloxycarbocyanine perchlorate (DiO)

was used to co-stain TAMRA-labeled DNA-lipid-P in HepG2 cells. As shown in Supplementary Figure 12, most TAMRA-labeled DNA-lipid-P still anchor on the cell membrane even after six hours of incubation at 37 °C. This phenomenon indicates that DNA-lipid-P anchors on cell membrane, not the membrane of organelles.

2. The authors should provide the measured ALP activity on HepG2 and U-2OS cells.

Response: Understand the reviewer's concern. According to the reviewer's suggestion, ALP activity of HepG2 and U-2 OS was measured by using of commercial-available Alkaline Phosphatase Assay Kit (Beyotime Biotechnology). As shown in Figure 5b and 5c, the ALP activity of HepG2 cells is 1205.5 nmol/min/mg protein which is 145-fold higher than that of U-2 OS cells (8.284 nmol/min/mg protein).

3. Is the DNA-lipid-P be stable in cellular environment? Will it be degraded by nuclease? Discussion is needed on this point.

Response: Understand the reviewer's concern. Actually, the stability of conventional oligonucleotides in complex biological environments is one of the important barriers against nucleic acids for further biological applications. However, many commercial-available chemical approaches have been reported to improve the stability of oligonucleotides against enzymatic degradation. For example, 2'-F and 2'-OCH₃ modification; phosphorothioate modification; and the reverse thymidine base at 3'-terminus (<http://www.glenresearch.com>). Additionally, in this work, cellular incubation was performed in buffer solution and DNA-lipid-P was anchored on the external surface of cell membrane. Therefore, DNA-lipid-P is stable enough in the environment; and, if necessary, chemical modification of DNA-lipid-P can be used for better resistance against enzyme degradation.

Reviewer #2 (Remarks to the Author):

In this manuscript, the authors reported a phosphorylated lipid-conjugated oligonucleotide (DNA-lipid-P) for ALP-dependent cell membrane anchorage. DNA-lipid-P with poor hydrophobicity shows very weak interaction with cell membrane in the absence of ALP. However, in the presence of ALP, hydrophilic DNA-lipid-P can be converted to hydrophobic DNA-lipid after enzymatic cleavage, resulting in strong fluorescence in the membrane of ALP highly expressed cancer cells. The novelty of this work is high and the manuscript is well prepared. The authors also provide a facile route toward the functionalization and a model to improve the properties of conventional lipid-conjugated oligonucleotides. This work is recommended for publication in Nature Communications after careful revisions.

1) "tcell" in Page 4 Line 1 should be checked out and revised.

Response: Thanks for the careful review. In the re-submission, "tcell" was revised to "the cell". Please refer to Manuscript, Page 4 Paragraph 1.

2) Compound characterization: The coupling constant should be given in $^1\text{H-NMR}$ data in the synthesis part. The H atoms of $^1\text{H-NMR}$ data (Supplementary Figure 35) of lipid phosphoramidite are not correct and should be carefully checked out.

Response: Thanks for the sincerely suggestion. In the re-submission, the coupling constant was added in $^1\text{H-NMR}$ data. Please refer to the synthesis part in the Manuscript (Page 11 and 12) and Supplementary Information (page 3 and 4). The $^1\text{H-NMR}$ data of lipid phosphoramidite has been revised. Please refer to (Supplementary Figure 40)

3) For better understanding, the chemical structures of TAMRA and Dabcyl should be added in the Supplementary Information.

Response: Thanks for the sincerely suggestion. In the re-submission, the chemical structures of TAMRA, Dabcyl and chemical phosphorylation reagent were added in Supplementary materials. Please refer to Supplementary Figure 1.

4) I suggest the authors to carry out co-stain experiment with commercial membrane probe to directly verify the location of the probe DNA-lipid-P.

Response: Understand the reviewer's concern and thanks for the sincerely review. According to the reviewer's suggestion, cellular colocalization between TAMRA-labeled DNA-lipid-P and commercial cell membrane probe (DiO) was performed. As shown in Supplementary Figure 12, most TAMRA-labeled DNA-lipid-P overlay with DiO, suggesting DNA-lipid-P had indeed anchored on the surface of cell membrane.

5) Direct imaging data should also be added to evaluate the phenomenon that oligonucleotides anchor on the external surface of cell membrane by using

Dabcyl-labeled complementary DNA.

Response: Thanks for the sincerely suggestion. According to the reviewer's suggestion, confocal fluorescence microscopy imaging of HepG2 cells treated with TAMRA-labeled DNA-lipid-P before and after hybridization with Dabcyl-cDNA were performed. Results in Supplementary Figure 15 indicate that the addition of Dabcyl-cDNA indeed quenches the fluorescence of cell membrane-anchored DNA-lipid-P, providing a direct imaging proof that DNA-lipid-P anchored on the external surface of cell membrane.

6) The authors should use TAMRA not Nile Red (Supplementary Figure 13) to demonstrate that neither DNA-lipid nor DNA-lipid-P is able to aggregate into micellar nanoparticles in buffer solution. Direct evidence like Dynamic Light Scattering (DLS) data should be also provided.

Response: Understand the reviewer's concern. In our re-submission, agarose gel electrophoresis was used to evaluate whether DNA-lipid or DNA-lipid-P assemble into aggregated state in buffer solution. As shown in Supplementary Figure 16b, 5 μ M TAMRA-labeled DNA, DNA-lipid and DNA-lipid-P (20 μ L) share similar migration rates in agarose gel, suggesting neither DNA-lipid nor DNA-lipid-P have appreciable self-assembly in buffer solution. Additionally, result of Dynamic Light Scattering (DLS) assay also support the conclusion that neither TAMRA-labeled DNA-lipid nor TAMRA-labeled DNA-lipid-P (10 μ M) have appreciable self-assembly in buffer solution (Supplementary Figure 16c).

7) Since elevated ALP level is a critical index of some diseases and even cancers, so if the authors could provide further imaging data with normal cell line which shows low ALP expression, I believe the readers can have a comprehensive appreciation of this work

Response: Understand the referee's concern and thanks for the careful review. Indeed, the selectivity anchorage of DNA-lipid-P on the cell membrane of cancer cells, not normal cells, could attract the readers. Previous literatures also reported that phosphate conjugate of tyrosine and cholesterol (pTC) has very low cell cytotoxicity to normal stromal cell line (HS-5), which expresses relatively low level of ALP; but higher cytotoxicity to cancer cells, which expresses elevated ALP (Bing Xu et al. J. Am. Chem. Soc. 2018, 140, 9566-9573). Herein, because of the problem of the time cost in purchasing the corresponding cells, we didn't perform the comparison of anchorage of DNA-lipid-P between HepG2 and HS-5. This study will carry on in the follow-up work.

8) The abbreviation for "Ange. Chem., Int. Ed." (Ref. 6, 16, 17 and 19) is not correct, please revise that. The name of the journal in Ref. 22 should also be checked carefully.

Response: Thanks for the careful review. In our re-submission, the mistake of abbreviation of "Ange. Chem., Int. Ed." (Ref. 6, 16, 17 and 19) has been revised to the correct type "Angew. Chem. Int. Ed." The name of the journal in Ref. 22 has been

revised to the correct name (Experientia). Please refer to the Reference section.

REVIEWERS' COMMENTS:

Reviewer #1 (Remarks to the Author):

The authors have addressed my previous concern. I support the acceptance of this excellent work.

Reviewer #2 (Remarks to the Author):

The authors have adequately addressed my previous comments and suggestions. The revisions are satisfactory and the changes are acceptable. The quality of the manuscript has been improved after revision. I do not have further criticism of the work.

Reviewers' comments:**Reviewer #1** (Remarks to the Author):

The authors have addressed my previous concern. I support the acceptance of this excellent work.

Response: We thank the reviewer for appreciating the work and recommending the publication in Nature Communication.

Reviewer #2 (Remarks to the Author):

The authors have adequately addressed my previous comments and suggestions. The revisions are satisfactory and the changes are acceptable. The quality of the manuscript has been improved after revision. I do not have further criticism of the work.

Response: We thank the reviewer for appreciating the work and recommending the publication in Nature Communication.